# Encapsulation of Antihypertensive Peptides from Whey Proteins and Their Releasing in Gastrointestinal Conditions

**DOI:** 10.3390/biom9050164

**Published:** 2019-04-27

**Authors:** Yolanda Alvarado, Claudia Muro, Javier Illescas, María del Carmen Díaz, Francisco Riera

**Affiliations:** 1Tecnológico Nacional de México/Instituto Tecnológico de Toluca, Toluca C.P 52140, Mexico; yalvaradop@toluca.tecnm.mx (Y.A.); fillescasm@toluca.tecnm.mx (J.I.); mdiazn@toluca.tecnm.mx (M.d.C.D.); 2Departament of Chemical Engineering and Envioronmental Technology, University of Oviedo, C. P. 33006 Oviedo Asturias, Julián Clavería 8, Spain; far@uniovi.es

**Keywords:** antihypertensive peptides, encapsulation, composite materials, peptides releasing, gastrointestinal conditions

## Abstract

Antihypertensive peptide fraction from whey protein hydrolysate <3 kDa (measured as angiotensin-converting enzyme (ACE) activity %) was isolated and encapsulated into three composite materials: alginate–collagen, alginate Arabic gum, and alginate–gelatin. The release behavior of peptide fraction from capsules was analyzed according to the encapsulation material efficiency, the characteristics of the capsules, and the released antihypertensive peptides during gastrointestinal digestion. The highest encapsulation efficiency was found in capsules of alginate Arabic gum (95%). In this case, the released peptides incremented their ACE activity (85%) after the digestion process, with respect to the initial ACE activity (74%). Whey antihypertensive fraction revealed five peptide sequences; however, other amino acid sequences were released from digested capsules. Protein databases confirmed some antihypertensive sequences indicating the peptides content from β-Lg and α-La. Consequently, new peptides could be revealed from whey antihypertensive fraction.

## 1. Introduction

Milk protein hydrolysates and their peptides are considered active molecules with biological properties to control and prevent chronic diseases, such as hypertension, obesity, diabetes, cancer, and oxidative stress [1,2]. Their benefits have been extensively tested in different reports, which has promoted their application in the elaboration of different products. Specifically, whey peptides are used as ingredients in the preparation of foods with different health benefits and they are also utilized to increment the functional properties of different types of foods or to provide new characteristics to them [3]. However, their health benefits in vivo are strongly affected, because these substances present a short lifetime, little stability, and severe degradation in gastrointestinal conditions [4].

Clinical applications of milk peptides have also been limited due to variable or poor effects when administered orally. Some of the causes are their variable instability and their fast degradation within the gastrointestinal tract (GIT) by digestive enzymes. Therefore, the active molecules exert a partial health effect, and thus, their target stimulus is achieved also in limited form [5,6].

In addition, hydrolyzed milk proteins and peptides may cause products with characteristics not favorable for their use, such as the alteration of their sensorial properties, texture, and color. The origin of these problems is attributed to the low bioavailability of molecules, their bitter taste, their hygroscopicity, or their probability of interacting with the food matrix [7].

Currently, numerous studies have demonstrated that some of these limitations may be overcome by the encapsulation of active molecules. Some of the advantages of this method are: (i) to preserve and to extend the activity, as well as to protect from the harsh environments such as heat, moisture, air, light, and intestinal enzymes; (ii) to retain its shelf life and to avoid the alteration of their chemical structure during the processing of a product; and (iii) to control the delivery of the active substance to a specific site [4].

Advances on encapsulation technology have led to the develop of an ever-increasing number of composite materials to prepare different carriers for containing and releasing active substances, namely, hydrogels, emulsions, nanoparticles, microspheres, and liposomes. They have been reported as a vehicle for different food ingredients like pigments, vitamins, fatty acids, antioxidants, and minerals [8]. However, little information about the encapsulation of milky peptides and their releasing mechanisms can be found in the literature. Studies on this topic demand different aspects that would fulfill their application and commercialization. For this purpose, it is necessary to demonstrate that biological molecules can reach the gastric tract without further degradation and can be released in the form of peptide fragments with considerable activity to ensure its action and benefits in the organism.

Research on this subject were reported by Yu et al. [9], who studied the encapsulation of bovine-serum albumin (BSA) into particles based on materials such as chitosan, alginate, and pectin. In addition, Yang et al. [10] tested the effect of the encapsulation of whey protein hydrolysates in capsules of maltodextrin and β-ciclodextrin. On the other side, Kavianinia et al. [11] analyzed the encapsulation of BSA in chitosan microspheres, and Ma et al. [12] used microspheres of sodium alginate to encapsulate whey protein hydrolysates. The most important results from these reports confirmed that peptide encapsulation improves the stability of active molecules and reduces their bitter taste.

Recent studies have also revealed that whey peptides encapsulation on nanoliposome carriers of soya lecithin [13], microspheres of sodium alginate [14], and β-Lg derived peptides in Poly Lactic-co-Glycolic Acid (PLGA) nanoparticles [15] could help release active molecules and improve their stability and action. Furthermore, some reports, such as Gómez-Mascaraque et al. [16], Bokkihim et al. [17], and Giroux et al. [18] employed simulated gastric fluid conditions to analyze peptides release behavior. They found that encapsulation offers protection to the active molecules and mainly can control the release of peptides; however, these aspects depend on the type of material of the carrier, the preparation processing techniques, the chemistry of the peptides, and their size. 

Nonetheless, to have some evidence about the benefits on the encapsulation of active molecules is not sufficient and more research is needed to increase the amount of information on this topic. Essentially, it is necessary to expand research on the release of whey active molecules during the digestion process in order to provide valuable data on the efficiency of peptides encapsulation technology.

The present work describes the encapsulation of antihypertensive peptides derived from whey protein hydrolysate and analyzes the release of these peptides in in vitro conditions. 

The effects of encapsulation on the preservation of antihypertensive activity and controlled release of whey peptides were studied in model conditions of simulated gastrointestinal digestion. Composite biopolymers based on the sodium alginate matrix and filler materials, such as gelatin, Arabic gum, and collagen were tested as encapsulating materials.

## 2. Materials and Methods 

### 2.1. Materials

The following reagents were supplied by Merck (Darmstadt, Germany). Sodium alginate (SA) medium viscosity; Arabic gum (AG) from the acacia tree (branched polysaccharide); calcium L-lactate hydrate ≥98% (calc. based on dry substance, KT); *N*-Hippuryl-His-Leu hydrate powder, ≥98% HPLC grade; angiotensin converting enzyme from rabbit lung ≥2.0 units/mg protein; ethyl acetate anhydrous, 99.85%; potassium phosphate dibasic, reagent grade (K_2_HPO_4_). 

Gelatin (GE) alkaline-processed (type B) from bovine skin and collagen (CO) from calfskin were obtained from Colloidal Duche S.A. Toluca, México. Bovine cheese whey was kindly supplied by a dairy industry in Mexico and the protein concentration in whey was determined as 11.5 ± 0.5 g/L.

*Bacillus subtilis* (free living strain 05) was obtained from the Research Laboratory of Environmental Engineering from the Institute Technologic of Toluca, Toluca, México. One part of this strain was placed in 100 mL of whey milk (11 g/L of protein) and it was incubated for 24 h at 37 °C. The *Bacillus subtilis* culture was subsequently used in all experiments of whey protein hydrolysis.

### 2.2. Whey Protein Hydrolysis 

Whey protein concentrate (WPC), with 84% protein and 10% lactose, was obtained by ultrafiltration processes (UF) of bovine whey samples using cross-flow polymeric membranes (15 kDa cut-off and 0.020 m^2^ effective membrane area). 

WPC was hydrolyzed by direct fermentation using *B. subtilis* biomass. The hydrolysis processes were carried out in Erlenmeyer flasks. In each test, 100 mL of WPC were incubated with *B. subtilis* biomass (10% *v*/*v*; 2.6 × 10^8^ cfu/mL). 

The flasks were sealed and put under orbital stirring at 200 rpm at 50 °C, and then underwent hydrolysis for 6 h (Alvarado et al., 2018). Proteolytic action was stopped by separating the free cells. The obtained samples were centrifuged at 10,000 rpm for 5 min; afterwards, supernatants were separated and identified as whey protein concentrate hydrolysate (WPCH) products. WPCH products were frozen and stored at –10 °C until further analysis. 

### 2.3. Fractionation of Hydrolysate Product 

The WPCH product was first fractionated by a polymeric MICROZA membrane with a molecular weight cut-off (MWCO) of 3 kDa (Pall Corporation). The whey protein concentrate hydrolysate permeate (WPCHP) fractions <3 kDa and the whey protein concentrate hydrolysate retained (WPCHR) fractions were collected during the filtration. Subsequently, the WPCHP was fractionated (20 mg/mL) by the gel filtration chromatography technique, using a Waters Liquid Chromatography System (USA) with a dual pump (Waters model 1525) and a Sephadex G-25 column (2.5 cm × 70 cm). Fractions were collected at the flow rate of 0.6 mL/min and the absorbance, at 220 nm, was monitored. At last, each fraction was lyophilized and stored at –10 °C until further analysis.

### 2.4. Determination of ACE Inhibitory Activity of Whey Peptide Fractions

Angiotensin-converting enzyme (ACE) inhibitory action was determined in peptide fractions, according to the method of Cushman and Cheung [19], with some modifications. The following solutions were used in the ACE quantification: (i) Histidyl-hipuryl leucine (HHL) (substrate of ACE), was prepared by employing 54 mg dissolved in 25 mL of dibasic potassium phosphate buffer solution 0.1 M, and 0.3 M sodium chloride at pH 8.3, resulting in a final 5 mM concentration of HHL; (ii) ACE (0.5 mg) was dissolved in 1 mL of glycerol solution, 50% *v*/*v*; (iii) The sample solution (M) was prepared with 40 μL of peptides sample, 100 μL of HHL solution, and 40 μL of ACE solution. The control solution (C) was prepared mixing 40 μL of buffer solution, adding 100 μL of HHL solution, and 40 μL of ACE solution. Meanwhile, the blank solution (B) was prepared mixing 40μL of distilled water with 100 μL of HHL solution, and 40 μL of ACE solution. The obtained samples M, B, and C were centrifuged at 5000× *g* for 5 min, and incubated at 37 °C for 30 min. After this time, the ACE enzyme was inactivated with the addition of 150 μL of HCl 1 N. After, 1 mL of ethyl acetate was added in each solution (A, B, and C), shaken in a vortex, and centrifuged for 10 min at 1541× *g*; 750 μL of the supernatant (organic phase) were taken and evaporated by heating them at 85 °C for 15 min. Then, 2000 μL of distilled water were added to the residue with orbital stirring. Lastly, the absorbance, at 220 nm, was measured for all samples, using a UV–Vis spectrophotometer LAMBDA 35 Perkin Elmer (Waltham, Massachusetts USA).

Inhibition of ACE was expressed as a percentage of the residual ACE activity, which was calculated according to Equation (1).
ACE inhibitory activity (%) = (C−M) × 100/(C−B)(1)
where C is the absorbance of the control solution C; M is the absorbance in the sample; and B is the absorbance in the blank solution B. 

The IC_50_ value (the concentration of inhibitor resulting in a 50% reduction of ACE activity) was calculated by the regression analysis from the ACE inhibition curve obtained with increasing amounts of the inhibitor. The IC_50_ values were expressed as ACE%. 

### 2.5. Encapsulation of the Peptide Fractions

Fractions with the highest ACE inhibitory activity were encapsulated in composite matrices of sodium alginate (SA) with Arabic gum (AG), collagen (CO), and gelatin (GE) as the filler materials, and also used as individual complementary constituents.

The encapsulating carriers SA-AG, SA-GE, and SA-CO were prepared as follows: AG, CO, or GE was dispersed in a SA solution 1.5% (p/v) under stirring, followed by sonication for 15 min. After, the hydrolysate fraction with the highest ACE inhibition was dispersed in each matrix, in a weight ratio of 4:1 at 30 °C. Capsules with different mixtures between the matrix and fractions were obtained by the extrusion method and dropped into calcium lactate 5% wt. The obtained capsules were separated by decantation and washed with distilled water to eliminate the lactate excess. 

The morphology and porosity of the encapsulating matrices were determined by means of scanning electron microscopy (SEM), employing a JEOL JSM-6610LV microscope. Surface area, pore volume and pore-size distribution was evaluated by Brunauer, Emmett and Teller (BET) method, using a analyzer BELSORP-aqua3, BEL Japan, Inc.

The entrapment efficiency of whey peptides from f4 fraction in the encapsulated matrices SA-CO, SA-AG, and SA-GE was determined by dissolving 50 mg of capsules in 1 mL of methanol or an alkaline solution. This parameter was calculated as the encapsulation efficiency (%) and it was defined as the ratio between the protein content of non-encapsulated peptides and the released peptides from the dissolved capsules. Finally, the protein content of the filtrate was measured by the Biuret method.

### 2.6. Simulated Gastrointestinal Digestion of Encapsulated Peptide Fractions

Kinetics of the released peptides from encapsulation was analyzed through in vitro studies under simulated gastrointestinal conditions (SGC) according to the protocols of the digestive phases reported by Minekus et al. [20]—1) Simulated salivary fluid (oral phase, SSF), 2) simulated gastric fluid (gastric phase, SGF), and 3) simulated intestinal fluid (duodenal phase, SIF). 

The saliva fluid was obtained preparing a solution with 2.38 g Na_2_HPO4, 0.19 g KH_2_PO4, and 8 g NaCl dissolved in 1 L of distilled water. The pH was adjusted to pH 7 and α-Amylase (EC 3.2.1.1) was added to the mixture to obtain 200 U enzyme activity. Gastric fluid was obtained by mixing 0.31% of pepsin enzyme, EC 3.4.23.1 and 0.03 M NaCl. The pH value was adjusted to pH 1.5 with HCl 1 M. Intestinal fluid was prepared dissolving 0.05 g of pancreatin and 0.3 g of porcine bile extract in 35 mL of NaHCO_3_ 0.1 M. The pH was adjusted to pH 6 with NaHCO_3_ 0.1 M.

The in vitro gastrointestinal experiments were carried out in glass tubes, considering SSF, SGF, and SIF phases. In the SSF phase, 10 mL of saliva fluid were put in contact with encapsulated peptide fractions (50 capsules of each matrix) and free fractions equivalent to 4 mg/mL. Samples were incubated at 37 °C, at a pH 7 for 10 min, in a shaking incubator stirred at 150 rpm. Subsequently, the gastric phase was obtained from the digestion of gastric fluid solution (10 mL) and porcine pepsin (2000 U/mL). The mixture was incubated at 37 °C and pH 3 for 2 h in a shaking incubator at 150 rpm. The gastric digest fluid was mixed (7.5 mL) with porcine bile extract (10 mM) and porcine pancreatin (100 U/mL of trypsin activity), 1.25 mL of 120 mM NaCl, and 1.25 mL of 5 mM KCl. The pH of the mixture was adjusted to 7 with NaOH 1 M; the mixture was incubated for another 2 h at 37 °C, and stirred at 150 rpm. To finish the gastric process, in the last phase, the protease inhibitor Pefabloc^®^ (1 mM) was added. 

All of the digested products from each phase were centrifuged at 500× *g* for 15 min; then supernatants were collected and centrifuged at 10,000× *g* for 5 min for a further analysis. The non-digested capsules during the gastric process were suspended in acetic acid (20% *v*/*v*) or NaOH (20% v/v) to dissolve them. The ACE inhibitory activity, the concentration of proteins and the identification of amino acid sequences, released from both the encapsulated and the free peptides, were measured in each phase and matrix. The ACE was determined by the method described in Section 2.4, and the concentration of proteins was determined by the Biuret method. 

### 2.7. Analysis and Identification of Amino Acid Sequences from the Released Peptides in Simulated Gastrointestinal Digestion from the Encapsulated f4 Fraction

Samples of released peptides, after each phase of the gastrointestinal digestion, were exposed to sequence identification by liquid chromatography and mass spectrometry.

The obtainment and identification of peptides was performed using a (RP-HPLC), Agilent 1200 series Reverse Phase Liquid chromatography–mass spectrometry (LC/MS system), coupled to an Agilent 6500 Series Q-TOF LC/MS system, Quadrupole-Time-of-flight liquid. Samples (10 µL) were injected into a Zorbax Eclipse C18 (4.6 mm × 250 mm, 5 μm) column, as the eluting program two mobile phases were employed, water containing 0.1% trifluoric acetic acid (TFA) (phase A, 30 min) and acetonitrile 35% with 0.1% TFA (phase B, 30 min) at a flow rate of 2.0–2.5 mL/min, temperature of 95 °C and pressure of 550 bar.

The obtained eluents were injected into the MS system. The MS analysis was performed in the positive ion mode for Auto LC/MS between 100 and 2500 *m*/*z* range; and the target masses 1221 and 322, respectively. The major peaks were selected to identify the peptide sequences employing the protein databases LC/MS Chemstation Software and Agilent MassHunter Workstation Software-Qualitative Analysis (Santa Clara, CA, US) (detection limit range 0.01 mg/L), allowing the identification of the peptides. The used parameters for both programs followed the next sequence: selection of the peptide charge (1+, 2+, 3+); tolerances of 1.0–1.5 Da for both the precursor and fragments ions; and the use of trypsin as proteases. The obtained sequences were also compared with peptide from whey proteins from the universal protein database resource (UniProt database) [21].

### 2.8. Statistical Analysis

All determinations were performed in triplicate. The experimental data were evaluated using the analysis of variance, followed by Duncan’s new multiple range test. Statistical significance was used to evaluate the differences between means at a significance level of *p* < 0.05.

## 3. Results and Discussion

### 3.1. ACE Inhibitory Activity of Hydrolysate Fractions 

The hydrolysis of WPC after 6 h of proteolytic action of *B. subtilis* was confirmed by means of the hydrolysis degree (DH%). The hydrolysate product WPCH showed a DH range of 35%–40%. Moreover, WPCHP and WPCHR fractions from membrane filtration (3 kDa) showed protein concentrations of 4.18 mg/100 mL ± 0.8 and 1.61 ± 0.8 mg/100 mL, respectively. This means, that a higher content of short peptides <3 KDa were present in fractions WPCHP; meanwhile, 4 fractions, f1–f4 from WPCHP, were obtained as result of the gel filtration chromatography, which corresponded to fractions f1, from 50 to 80 min; f2, from 120 to 130 min; f3, from 200 to 210 min; and f4, from 220 to 250 min.

In the case of the ACE inhibition percentage (%) from f1 to f4, Figure 1 shows data on the biological activity of these fractions. From this figure, it is shown that f1–f4 exhibited a high ACE-inhibitory effect. The non-fractioned WPCHP presented 45.67 ± 1.2%; whereas fractions f1–f4 showed 56%–80% of ACE inhibition. It is noteworthy that the ACE inhibitory activity from f4 was the highest, exhibiting a value of 80.65 ± 1.5%. 

These results were attributed to WPC hydrolysis by the proteolytic action of serine protease and metalloprotease enzymes from *B. subtilis* [6], which resulted in a satisfactory release of ACE inhibitory peptides.

The highest ACE inhibitory activity from f4 was associated with the lowest molecular weight of the contained peptides, because of the presence of short peptide sequences, as well as the peptide structure with a specific amino acid composition.

Although several problems have been found for the identification of short peptides from hydrolyzed complexes [22], there is some information about the relationship between the short peptides and their ACE-inhibitory activity. Specifically, di- and tri-peptides are linked to ACE enzymes, because they are bonded to an active site [23,24]. 

The position of the amino acids within the sequence (at the N- or C-terminal) and its adjacent residues, have also been recognized as important factors that affect ACE activity from peptides. ACE shows affinity to peptide sequences containing amino acids in the C-terminal, such as proline and leucine, and hydrophobic amino acids (or positively charged) such as phenylalanine, tryptophan, and tyrosine. Wu et al. [25] also found that the residue of the C-terminal tetrapeptide may determine the potency of ACE inhibition—with preference for tyrosine and cysteine in the first C-terminal position; histidine, tryptophan and methionine in the second; leucine, valine and methionine in the third; and tryptophan in the fourth position. 

The occurrence of these amino acids in the released peptides from whey hydrolysis was currently confirmed by Alvarado et al. [6]. Whey protein produced from the hydrolysis process with *B. subtilis* enriched peptides with high content in amino acids residues as valine, leucine, tyrosine, phenylalanine, proline, arginine and isoleucine, and they were the principal contributors of ACE activity. Other obtained results on the production of ACE inhibitory peptides by enzymatic action of *B. subtilis* were similar to previous reports from protein hydrolysis from marine organisms [14,23].

### 3.2. Capsules Characteristics and Encapsulation Efficiency

The obtained capsules SA-CO, SA-AG, and SA-GE, loaded with whey peptide f4, seemed very similar and had a spherical form with a diameter size of 1–2 mm; however, pore dimensions and surface area presented important differences associated to the composite materials used in the encapsulation process. Moreover, these data were comparable with SA capsules. The obtained data are shown in Table 1.

The SA-CO, SA-AG, and SA-GE capsules incremented the surface area in comparison with the individual SA capsule. The highest value was found for the SA-GE capsules, followed by SA-CO, and finishing with SA-AG. Therefore, a greater contact between gastric fluids and SA-GE capsules could be expected; this could cause an important capsule erosion and the release of peptides in shorter times. The filler materials also modified pore diameter. SA capsules showed high pore diameter, whereas their composite presented a reduced pore. This modification possibly improved the mechanical properties and the peptides retention, and prevented peptides to leak from capsules, because capsules of SA have displayed fragility and friability with higher pore sizes, which may also cause some fractures and disintegration of the material during the hydrolysis process.

In regard to encapsulation efficiency, SA-CO, SA-AG, and SA-GE presented higher values, ca. 85%–95%. SA-AG capsules reached the highest efficiency, followed by SA-CO, and lastly, the SA-GE capsule. This result was associated with materials properties, such as their capacity of entrapment, their binding abilities and their interactions between peptides and f4 fraction. Specifically, AG is a highly branched heteropolymer consisting of sugars, which contains a small amount of protein, covalently linked to the carbohydrate chain. Its bind with SA (polysaccharide) could lead to low intermolecular interactions of f4 fraction; thus, the entrapment mechanism of SA-AG may be owed to its excellent film-forming property, providing a better trapping, and a much higher release ratio of f4 fraction. As for SA-CO and SA-GE capsules, the entrapment and release properties can be also attributed to their interactions with f4 fraction. Primarily, the obtained result with SA-GE capsules could be associated to the strong interactions with f4 fraction, since GE has a negative charge [26]. In addition, the formation of complexes due to the instability of f4 fraction, and its interactions with SA matrix, could be also considered in the obtained data. The interactions of f4 fraction with GE could also contribute to the transformation of f4 to other less stable forms, which also makes its releasing difficult, and thus lowers encapsulation efficiency. 

The obtained SEM micrographs of the capsule surface complemented the last result. Figure 2 shows micrographs of SA-AG, SA-CO, and SA-GE capsule surfaces loaded with f4 peptide fraction. 

Micrographs show surfaces from capsules without fissures and with some clusters of material, evidencing the adequate dispersion of fillers onto SA capsule; cracks and porous present in these images are characteristic of SA, due to the crosslinking process with calcium chloride. Particularly, the following differences were observed in the surface of the capsules.

SA-GE and SA-CO presented a more homogeneous surface without the presence of holes; whereas, SA-AG exhibited a heterogeneous surface with the possible formation of some pores in the structure, confirming the results of pore dimensions showed in Table 1 for each composite capsule.

It was also possible to observe the peptide f4 fraction on the capsules; a higher amount of peptides was evident in the surface area from SA-AG and SA-CO, indicating the homogeneous distribution of peptides throughout materials. Nevertheless, in SA-GE capsules, peptides were not visible; therefore, complex formations of f4 and GE could give a possible explanation to this result.

The principal applications of SA, CO, AG, and GE as delivery materials have been previously reported [26]. They have been often studied as individual carriers of active substances as drugs and food ingredients; properties of gastro retentive dosage and the augment of the gastric residence time into the delivery system have been confirmed in these reports [27,28].

Characteristics of dispersion, assembly, and biocompatibility of SA have also been tailored via composite preparation. The SA composites have showed structures more resistant and less porous, enhancing the retention and the protection of different biomolecules [7]. However, its application in synthesized SA-CO, SA-AG, and SA-GE capsules of whey peptides is little known in the literature. Therefore, the obtained information can be valuable in the area of production and encapsulation of whey peptides.

Previous works about whey peptides encapsulation have indicated the benefits that offer different carrier systems in the protection and releasing of peptides [4,7]. Mainly some polysaccharides and proteins as encapsulation materials (capsules and particles) have been utilized to inhibit the bitter taste and reduce the hygroscopicity of peptides [7]. However, it has also exposed that the encapsulating system types, encapsulating material, and peptides affect encapsulation efficiency. Peptides interaction and their molecular mass can cause low retention and releasing of the active peptides. Zhang et al. [28] tested the preparation of SA at different pH values (3, 5, and 7), which influences in the protein–alginate electrostatic interactions, and thus affects the release of peptides.

Other excellent carriers of whey protein hydrolysates were described in Kavianina et al. [11]. They utilized microparticles of maltodextrin or maltodextrin/β-cyclodextrin, enhancing the hygroscopic properties of whey peptides; the stability and activity was tested. In addition, these materials did not present interactions with peptides and they did not exhibit negative effects on their biological activity. Mohan et al. [13] tested soy lecithin-derived nanoliposomes as encapsulating systems of whey peptides to analyze the influence of the molecular weight range in peptides encapsulation efficiency. They showed that the different distribution of peptides between the polar surface/core and the hydrophobic bilayer of the liposomes affects the encapsulation efficiency of whey peptides.

### 3.3. Simulated Gastrointestinal Digestion of ACE-Inhibitory Peptide Fractions

Table 2 shows the results of the release kinetics of f4 peptide fraction during simulated gastric phases from 50 capsules (approximately 4 mg/L of protein and 78% of initial ACE). Data from gastric phases of non-encapsulated f4 peptide fraction is also presented as a control analysis. 

Given the condition of the non-encapsulated f4, its digestion showed the highest release of peptides during the oral phase, with a small ACE inhibition percentage. Meanwhile, its digest phase revealed a lower amount of released peptides, as well as a reduced ACE activity, exposing the probable production of inactive peptides by enzymatic degradation in gastric fluid, as it was expected. The instability and fast degradation of free peptide fractions have been exposed in previous reports [1,5].

The digestion of encapsulated f4 fraction displayed different results. An optimal dose and a controlled release were mainly obtained in the case of SA-CO and SA-AG capsules, since peptides were released during the three phases of digestion. The highest ratio of the released peptides and ACE inhibition percentage were found as follows: gastric phase > oral phase > duodenal phase.

Specifically, peptides from SA-AG capsules incremented their ACE activity (85%) after digestion, with respect to the initial ACE activity (74%); the optimal release and hydrolysis of f4 fraction can be attributed to this result. On the other hand, ACE activity from peptides from SA-CO and SA-AG capsules, 50% and 70% respectively, was lower than SA-GE; however, their activity was not comparable with ACE activity from peptides from the non-encapsulated f4 fraction (<20%), revealing for each encapsulated material, an effective protection and a higher release efficiency of f4.

The degradation details on the behavior of these capsules during the digestive process are described in Figure 3. A primary degradation of the encapsulated filler materials was detected in the gastric step; capsules erosion with filler materials CO, AG, and GE were observed during the gastrointestinal phase; whereas, SA capsules did not suffer any change. Thus, the release of peptides was attributed to the diffusion of f4 fraction, as well as materials erosion and dissolution. Hence, the contained peptides corresponded to both, f4 and products of their hydrolysis or degradation by gastric enzymes and pH conditions. This fact could explain the differences of ACE inhibition percentage from the capsules; peptides from SA-CO and SA-GE exhibited a lower ACE % than SA-AG. The reduced ACE % was associated to inactive peptides, due to digestive enzyme attack and to the release of degraded amino acid sequences. The structure of f4 fraction could also suffer some change from its original form, releasing inactive peptide fractions. In addition, the formation of complexes, predominantly f4-GE, could explain the low ACE activity of peptides from SA-GE. 

In the case of the oral phase, the released peptides could be refereed to peptides diffused from f4 fraction, contained in the capsule surface; in addition, a possible slight degradation of fillers could also be presented. Meanwhile, the presence of peptides in the duodenal phase could be attributed to the existence of f4 fraction, peptides from f4, and degraded CO or GE and their peptides with lower activity.

At this point, data from the digestion of f4 fraction contained in SA-GE capsules are also remarkable. In this case, the ACE inhibition % was lower than the released peptide sequences from SA-CO and SA-AG capsules. This result was linked to the strong interactions between peptides from f4 fraction and SA-GE, still in the duodenal phase; assuming the formation of complexes f4-GE and thus, a reduced hydrolysis and releasing of active peptides, as well as, possible reactions of f4 with the carboxylic acid groups of the SA, which may result in protein denaturation. 

In the case of the digestion of SA-AG capsules, the release of a greater amount of ACE active peptides was associated to the integral retention of f4 and its optimal release in the gastric phase to produce active amino acid sequences. 

### 3.4. Amino Acid Sequences from Released Peptides in Simulated Gastrointestinal Digestion from Encapsulated f4 Fraction

Relevant information on released amino acid sequences from f4 (non-encapsulated and encapsulated samples) was obtained before and after their digestion. 

In order to illustrate these results, Figure 4 shows images of MS/MS fragmentation spectrum of the studied samples as (A) a fraction from undigested and non-encapsulated f4; (B) an obtained fraction from a digested SA-AG capsules, which corresponds to the gastric phase; (C) an obtained fraction from a digested SA-AG from the duodenal phase.

Note that Figure 4A shows the spectrum of a fraction with an approximate charge on *m*/*z* = 1195, with a prominent ion at *m*/*z* = 717.35, an intensity of 1.6 × 10^4^ and with two generated fragments—considering peptides counts for intensities above 5000 units for their identification. Figure 4B displays a similar spectrum of the same fraction (*m*/*z* = 1195). In this case, an identical prominent ion is observed at 717.36; but it is noteworthy that the intensity of the base peak was greater around 4.2 × 10^4^; thus, the mass region of the spectrum was normalized, according to this ion. Consequently, the released fragments were similar to the spectrum of Figure 4A; however, another ion fragment at *m*/*z* = 429.31 could be observed. This was an indication of the different f4 sequences obtained after the digestion of SA-AG capsules. Meanwhile, Figure 4C shows the spectrum of the same fraction; however, two prominent ions were found at *m*/*z* = 619.38 and 371.22 (above 5000 units of intensity). It is worth mentioning the presence of the recognized ion (*m*/*z* = 717.34), but with a lower intensity value. These results are probably attributed to the digested fraction in the duodenal phase.

Sequences of released fragments from samples relatives to f4 are presented in Table 3. The data include amino acid sequences from digest phases of non-encapsulated and encapsulated f4 into SA-CO, SA-AG, and SA-GE capsules in simulated conditions. Data also contains undigested free (non-encapsulated) sample f4 and digested free f4 as data control. According to MS/MS fragmentation of f4, some fractions matched the protein databases. Identified amino acid sequences from f4 peptides and their protein origin are presented according to the data from MS spectra against UniProt database.

As it can be observed, Table 3 shows that the identified sequences from f4 corresponded to residues of β-Lg and α-La proteins, which is consistent with the previous work performed by Alvarado et al. [6], where it was demonstrated that the hydrolysis of WPC by *B. subtilis* for 6 h, produced hydrolysates with high content of residues of these proteins. Other reports such as Kobayashi et al. [29], and Cicero and Borghi [30] also revealed that α-La and β-Lg are the major sources of ACE-inhibitory peptides, and their activity may be comparable to that of captopril, an antihypertensive drug.

Table 3 also exposed that undigested and non-encapsulated f4 sample (first sample control, the more potent ACE inhibition fraction obtained from WPCH) showed five amino acid sequences, all containing more than one hydrophobic and aromatics amino acid, which were identified as promoters of ACE inhibitory activity; such as valine (V), glycine (G), and proline (P). Especially, fragments 1 and 5 contain V and P in the first and last sequence position. Positive charged amino acid, such as arginine (A) and lysine (L) at the C-terminal were also found in these structures (fractions 1, 3, and 5); they are also known as amino acid contributors of ACE inhibitory or antihypertensive effect [6]. Meanwhile the digestion of non-encapsulated f4 (second sample control) showed other amino acid residues. In this case, the gastric phase revealed three peptide sequences; displaying the effect of enzymatic action of the pepsin and the pH of gastric conditions in the degradation of precursors sequences from undigested f4. In addition, it is also possible to observe that no fragments were found in the duodenal phase; this result was associated to the complete degradation of these sequences in duodenal conditions, as was expected.

In the case of the digestion of encapsulated f4, the presence of fragments after this process indicated that capsules were effective carriers, protectors, and release agents; this result was particularly outstanding in SA-AG and SA-CO capsules. Clearly, SA-AG obtained the highest amount of released peptides in both, gastric and duodenal phases, showing 4 peptide sequences in each phase respectively, but only 4 sequences were confirmed by matching the protein databases.

Other important results were obtained from SA-CO and SA-GE capsules. SA-CO showed three and two peptide sequences in the gastrointestinal and duodenal phase respectively. Meanwhile, SA-GE exposed three peptide sequences in each stage. However, only two sequences from SA-CO and SA- GE were identified in protein databases. 

Summarizing, simulated digestion of encapsulated fraction f4 in SA-CO, SA-AG, and SA-GE exposed 19 peptide sequences. In parallel, only 11 peptide sequences were found in protein database and de novo searches.

Under these results, new peptides from whey f4 samples could be claimed. The digestion of f4 causes the releasing of peptides, showing new amino acid sequences. However, more studies are required to confirm the existence of novel peptides and also to define which of them are ultimately responsible for the potent antihypertensive activity.

The difference in released sequences from f4 during capsules digestion were attributed to composite material properties SA-AG, SA-CO, and SA-GE; material interactions with peptides fragments; the formation of new peptides by hydrolysis of f4 during gastric and duodenal phase, and the formation of complex mixtures of peptides, due to hydrolysis of CO and GE, during the digestion of these capsules. Consequently, it was recognized, that only some digested fragments were derived from the precursors sequences from f4, indicating changes in the structure of the initial sequences from f4. 

The obtained data were associated to different aspects that affected sequences isolation and identification during the chromatographic analysis, such as formation of complex mixtures of peptides, difficulty in sequencing peptides, and peptide fragment interactions with CO and GE which affected peptide recovery. Possible peptide degradation and the formation of new and different fragments could also be exposed, due to CO and GE that have a protein origin. 

Previous reports have also established that GE particles show complex chromatograms after the digested samples precluding the identification of peptides from the hydrolysate [16]. Their results also exposed that the gelatin carrier released 58 peptides from encapsulated whey active fractions; however, after particles disruption, 13 peptides were not identified, associating this result to peptide–matrix interactions. 

Comparable active sequences from whey proteins have also been described in other studies. For example, Tavares et al. [31] identified three novel peptides with ACE-inhibitory effect, corresponding to α-La f(16–26), α-La f(97–104), and β-Lg f(33–42)—with KGYGGVSLPEW, KVGINYW, and DAQSAPLRVY sequences, each one showing IC_50_ values like 0.80, 25.2, and 13.0 µg/mL. On the other side, Ibrahim et al. [32] identified one potent ACE inhibitory peptide from β-Lg (residues 113–122) with PEQSLACQCL sequence, exhibiting IC_50_ values of 4.45 μM. 

## 4. Conclusions

In this study, antihypertensive peptide fraction (f4) with 74% of ACE inhibitory activity from whey protein hydrolysate was encapsulated using composite materials SA-CO, SA-AG, and SA-GE as capsules, in order to extend peptides activity and to control its release in the gastrointestinal phase. 

Besides, the studied capsules presented a high value (85%–95%) of encapsulation efficiency; from them, SA-AG achieved the highest level, followed by SA-CO and SA-GE. 

The digestion of encapsulate peptides revealed an optimal dose and release control; it was mainly obtained by SA-AG and SA-CO capsules, since peptides were released during the three phases of digestion. The highest ratio of released peptides and ACE inhibition percentage (%) was found as follows: gastric phase > oral phase > duodenal phase. In addition, peptides from SA-AG incremented their ACE activity (up to 85%), from their initial ACE activity; whereas, peptides from SA-CO and SA-GE (50%–70%) was lower than SA-AG. The reduced ACE % of peptides from SA-GE was associated to the inactive peptides, due to the digestive enzyme attack and to the release of degraded amino acid sequences. The structure of f4 could also suffer some changes from its original form, releasing inactive peptide fractions. The last results depended on materials properties, such as capacity of entrapment, binding abilities, and interactions with peptide fractions.

In the case of sequences identification, non-encapsulated and undigested antihypertensive peptides revealed five peptide sequences (peptide > 5000 intensity units); while, encapsulated peptides showed different sequences at the end of digestion phases; however, only some sequences were confirmed by matching the protein databases; consequently, new peptides could be obtained from whey fraction. Nonetheless, more studies are required to claim these peptides and also to define which of them are ultimately responsible for the potent antihypertensive activity. 

The obtainment of cleaner spectrums was also vital to get a successful way a sequencing de novo. Similarly, more chromatographic studies are required to obtain precise results about peptide sequences; the challenge to match a unique peptide sequence to the targeted molecular mass. 

Additional experiments are similarly essential to declare that the encapsulation procedure did not affect the initial fraction, since complex mixtures formation and peptide fragment interactions with CO and GE could affect peptide recovery. Possible peptide degradation and the formation of new and different fragments could also be discovered, since CO and GE are of protein origin. 

Another challenge was the test validating the effect of these novel peptides, because they should be verified in vivo. In addition, studies about peptides with <5000 units of intensity could also be exposed to explain the ACE-inhibitory effects from f4 samples, thus complementing this research. Additional experiments are similarly essential to declare that the encapsulation procedure did not affect the initial fractions from f4. 

## Figures and Tables

**Figure 1 biomolecules-09-00164-f001:**
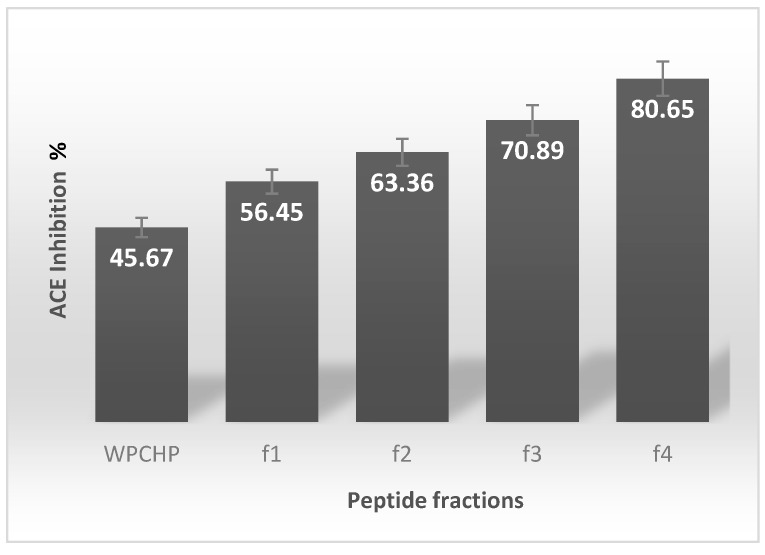
Angiotensin-Converting Enzyme inhibition % of whey protein concentrate hydrolysate permeate (WPCHP) and peptide fractions (f1–f4) from gel filtration chromatography. Fractions correspond as follows: f1, from 50 to 120 min; f2, from 120 to 180 min; f3, from 200 to 210 min; and f4, from 220 to 250 min. Values are expressed as the average ± SD of three independent determinations.

**Figure 2 biomolecules-09-00164-f002:**
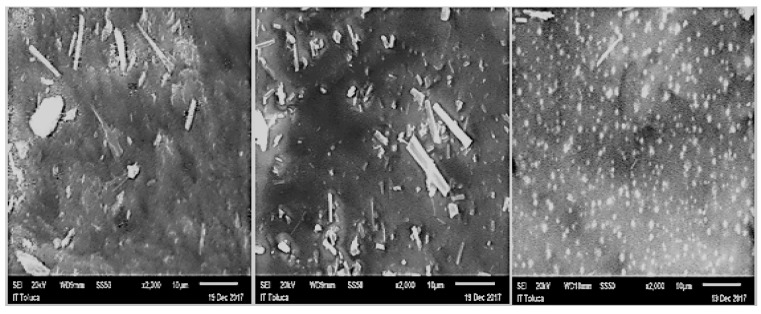
Micrographs from Scanning Electron Microscopy (SEM) of alginate-Collagen (SA-CO), Alginate-Arabic gum (SA-AG) and Alginate-Gelatin (SA-GE) capsules surface; from left to right, loaded with f4 peptide fraction.

**Figure 3 biomolecules-09-00164-f003:**
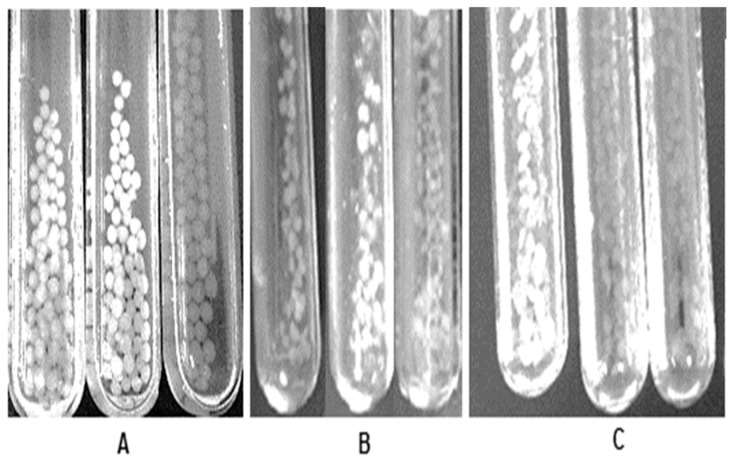
Images of f4 capsules during digest phases: (**A**) SA-AG, (**B**) SA-CO, and (**C**) SA-GE. From left to right: initial capsules, at the end of oral phase, at the end of gastric phase.

**Figure 4 biomolecules-09-00164-f004:**
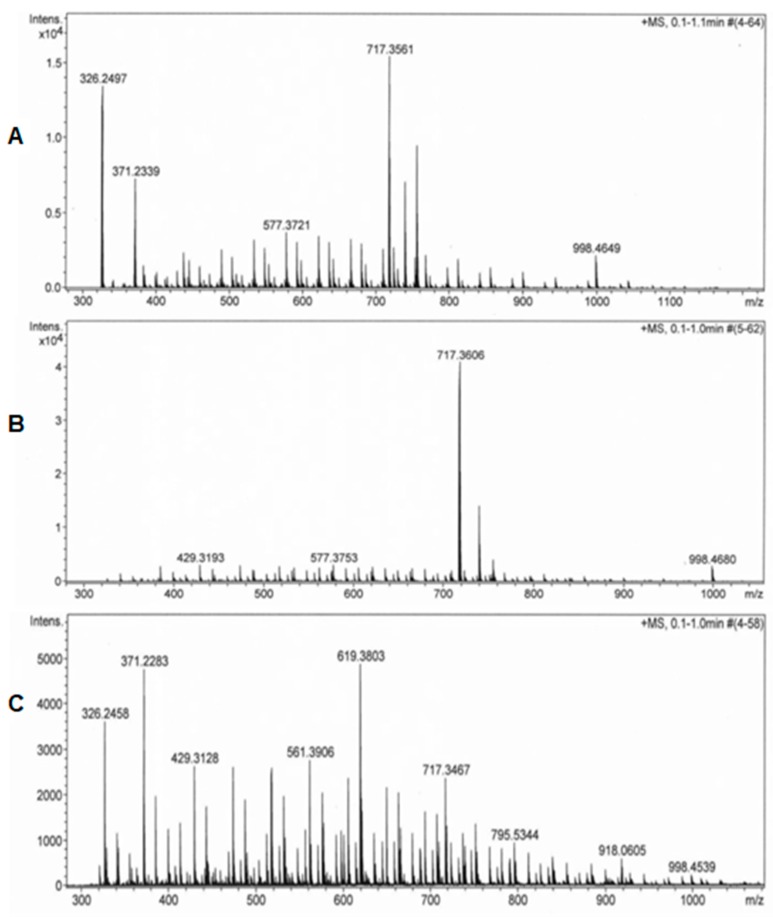
Mass/Mass (MS/MS) fragmentation spectrum of the studied samples as (**A**) a fraction from an undigested and non-encapsulated f4; (**B**) an obtained fraction from the digested SA-AG capsules corresponding to the gastric phase; (**C**) an obtained fraction from the digested SA-AG capsules from the duodenal phase.

**Table 1 biomolecules-09-00164-t001:** Capsules pore dimensions and encapsulation efficiency.

Capsules	Pore Volume (cm^3^/g)	Surface Area (m^2^/g)	Pore Diameter (nm)	Encapsulation Efficiency %
SA	0.0252	1.003	6.9	70
SA-CO	0.0019	1.405	5.5	90
SA-AG	0.0016	1.082	6.1	95
SA-GE	0.0018	1.95	3.8	80

**Table 2 biomolecules-09-00164-t002:** Release kinetics of gastric phases of f4 peptide fraction from capsules SA-CO, SA-AG, SA-GEm and non-encapsulated peptides.

Gastric Phase	Time (min)	Ratio of Released Peptides (mg/L)				ACE Inhibition (%)			
		Loaded Capsules of f4			Non-encapsulated f4	Loaded Capsules of f4			Non-encapsulated f4
		SA-CO	SA-AG	SA-GE		SA-CO	SA-AG	SA-GE	
Oral	10	1.5	0.5	1.0	2.3	25.3	9.6	18.6	15.2
Gastric	120	2.1	2.7	1.6	0.8	39.5	57.4	30.3	2.7
Duodenal	120	0.3	0.6	1.4	0.1	9.5	20.4	5.0	0.0
Total	250	3.9	3.8	4.0	3.8	74.3	86.4	53.9	17.9

**Table 3 biomolecules-09-00164-t003:** Released fragment and identified sequences from fraction f4 during its digestion in simulated conditions.

Sample Fraction f4	Digest Phase	Released Fragments/Suggested Sequence	Identified Sequence	Experimental Mass (Da)	Theoretical Mass (Da)	Protein Origin
Non-encapsulated	Undigested	1. VNLSMYNGIAL 2. ITPAVQMN 3.TVVSAPNYTLR 4. VAGTWY 5.LMTGYPVILYP	VAGTWY	695.4	695.2	β-Lg
Non-encapsulated	Gastric	1.SAPLR 2.GTW 3. TYV	SAPLR	543.6	543.2	β-Lg
Capsules SA-CO	Gastric	1. VLDTDYK 2. PAVQM 3. TSGYPV	VLDTDYK	852.6	852.4	β-Lg
	Duodenal	1.VDY 2.KIDAL	KIDAL	558.1	558.3	β-Lg
Capsules SA-AG	Gastric	1. ENSAEP 2. IPAVFK 3. VAGTWY 4. VSYT	ENSAEP IPAVFK VAGTWY	645.5 673.7 695.4	645.3 673.4 695.2	β-Lg
	Duodenal	1. VYT 2. AMAASDE 3. IPAVF 4. MDSF	IPAVF	545.5	545.3	β-Lg
Capsules SA-GE	Gastric	1.DGFYELYAME 2.VKTYMNATIK 3. KIPAVF	KIPAVF	673.1	673.4	β-Lg
	Duodenal	1. SAAGYT 2. DFELY 3. HTSGY	HTSGY	563.2	563.2	α-La

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
