# Peer review of "Encapsulation of Antihypertensive Peptides from Whey Proteins and Their Releasing in Gastrointestinal Conditions"

_biomolecules, 2019, doi:10.3390/biom9050164_

Round 1

Reviewer 1 Report

The work demonstrates that some whey hydrolyzate peptides have ACE inhibiting activity and that this activity is kept after their encapsulation and release into the intestinal content. The study is complex and uses up-to-date methods. However, all this research is pointless because these peptides have anyway no chance to be absorbed from the GIT and consequently  act in the body as ACE inhibitors and antihypertensive drugs. This is due to their high molecular mass and high hydrophilicity and partly also to their low resistance against enzymatic hydrolysis, although some hydrolytic stability has been demonstrated. So what is the purpose for filling the capsules with these peptides? 

Author Response

Reviewer #1: Observations

The work demonstrates that some whey hydrolyzate peptides have ACE inhibiting activity and that this activity is kept after their encapsulation and release into the intestinal content. The study is complex and uses up-to-date methods. However, all this research is pointless because these peptides have anyway no chance to be absorbed from the GIT and consequently act in the body as ACE inhibitors and antihypertensive drugs. This is due to their high molecular mass and high hydrophilicity and partly also to their low resistance against enzymatic hydrolysis, although some hydrolytic stability has been demonstrated. So what is the purpose for filling the capsules with these peptides?

Response.

Whey proteins are known to be precursors of peptides with different biologic activities. Whey protein hydrolysis leads the releasing of peptides with biologic action, as the antihypertensive activity. However, the peptides use as active molecule in food ingredients or clinical applications is limited, because the peptides must retain their biologic activity, survive gastrointestinal digestion and reach their target sites after absorption.

The digestive proteases and pH conditions in the gastrointestinal tract alter the activity of the peptides. Proteolytic enzymes can lead to peptide degradation and consequently loss of activity. Peptides with high basic amino acid content and long-chain bioactive peptides, especially ones with more than five amino acid residues, have lower tolerance to gastrointestinal digestion. In addition whey peptides present poor permeation when they are administered orally.

Actually, there are several reports about these topics.

Therefore, to exert their physiological effects in the organism, the peptides need to be protected against gastrointestinal degradation, and their permeation should be enhanced.  

The encapsulation is also an alternative of peptides protection and permeation and absorption enhance. The encapsulation systems offer protection from enzymatic degradation and facilitate uptake via cells. They are principally used for the preservation of biologically active from peptides.

Reviewer 2 Report

The paper deals with the encapsulation of an antihypertensive products obtained from whey protein hydrolysates. Antihypertensive peptide from hydrolysate of whey protein encapsulated using SA-CO, SA, AG and SA-GE as matrices material, were used to prolong the peptides activity and control its releasing during gastrointestinal phase. This study provides some new information about.

I have some comments.

Material and methods:

Pag. 3, Line 119: I suggest to write “Determination of ACE inhibitory activity of whey peptide fraction”.

Pag. 3, Line 95-98: Which was the concentration (log cfu/mL) of Bacillus subtilis culture? Please, specify.

Pag. 3, Line 123-124: Please, insert the brackets correctly.

Pag. 3, Line 127: Please, specify the acronyms the first time that you use. Please, specify the word “ECA”

Pag. 5, Line 201: What were the detection limits for the separated substances?

Results and discussion:

Some parts of this chapter are poor and only summarize the data presented in tables and figures.

Pag. 5, Line 217-218: of 4.18 mg/100 217 mL±0.8 and 1.61 ± 0.8 mg/100 mL - the record should be the same.

Figure 1.: Where are the different letters on fig 1 that indicate significant differences.

Pag. 6, Line 237-238: You should avoid to compare the results from other researches but make a more concise and constructive discussion.

Pag. 8, Line 300-304: Compare own results with other studies is important to draw the conclusions. In this case, I think you should avoid to cite the results of other papers in your discussion. Why did you compare your results with Zhang et al. Did you carry out your research at a particular pH?

Conclusions:

You should be more concise. The conclusions should not be a repetition of Results and Discussion.

Author Response

Reviewer 2

Material and methods:

Pag. 3, Line 119: I suggest to write “Determination of ACE inhibitory activity of whey peptide fraction”.

Response: According to the suggestion, subtitle was changed in the manuscript. Line 118

Pag. 3, Line 95-98: Which was the concentration (log cfu/mL) of Bacillus subtilis culture? Please, specify.

Response: Concentration of B. subtilis was specified in the manuscript. Line 104.

Pag. 3, Line 123-124: Please, insert the brackets correctly.

Response:  The brackets are were not incorrect, however that paragraph was rewritten for better understanding. Line 120-135.

Pag. 3, Line 127: Please, specify the acronyms the first time that you use. Please, specify the word “ECA”

Response: The word ECA was changed by ACE. Line 121, 130.

Pag. 5, Line 201: What were the detection limits for the separated substances?

Response: Detection limits for separated substances are specified in line 209.

Results and discussion:

Some parts of this chapter are poor and only summarize the data presented in tables and figures.

Pag. 5, Line 217-218: of 4.18 mg/100 217 mL±0.8 and 1.61 ± 0.8 mg/100 mL - the record should be the same.

Response: The records of peptides mass does not exactly have to be the same, depends on the membrane and the hydrolysates, because WPHP are peptide fractions < 3 kDa, and WPHR are peptides > 3 kDa.  In this report, a major record of peptides was obtained in WPCHP, indicating that during the hydrolysis were principally obtained peptides <3 kDa. Thus, the established values are corrects. 

Figure 1. Where are the different letters on fig 1 that indicate significant differences?

Response:  Fig 1, was corrected; the mentioned paragraph was a mistake.  

Pag. 6, Line 237-238: You should avoid to compare the results from other researches but make a more concise and constructive discussion.

Response: line 235-255 present the results discussion about to peptides fraction f1-f4 contained.  

Pag. 8, Line 300-304: Compare own results with other studies is important to draw the conclusions. In this case, I think you should avoid to cite the results of other papers in your discussion. Why did you compare your results with Zhang et al. Did you carry out your research at a particular pH?

Response:  Authors consider that is important to compare the obtained results with previous studies to show the direction and differences between each report, and where each one contribute to known about this topic. Line 322-339 were changed to indicate the contribution of previous works.

Zhang e t al., is cited in this report, because they utilized SA material to encapsulate whey peptides, and they demonstrated that the preparation of SA affect the peptides interaction and peptides releasing. Line 328-330.     

Conclusions:

You should be more concise. The conclusions should not be a repetition of Results and Discussion.

Response: The conclusions paragraph was corrected according the suggestions. 

Round 2

Reviewer 1 Report

I still mean that a study of releasing of antihypertensively active  peptides from capsule inside the GIT has no practical meaning because such peptides cannot be absorbed due to their big molecular mass and hydrophility. A serching for new antihypertensine petides can be performed by much more simple laboratory methods without specially prepared capsules.

Author Response

Reviewer 1

I still mean that a study of releasing of antihypertensive active peptides from capsule inside the GIT has no practical meaning because such peptides cannot be absorbed due to their big molecular mass and hydrophilicity. A searching for new antihypertensive peptides can be performed by much more simple laboratory methods without specially prepared capsules.

Response:

The antihypertensive effects of whey peptides have been demonstrated in several studies in vitro and in vivo. The results from experimental hypertensive rats (SHR) and extensive clinical data for human subjects have shown that that several ACE inhibitory peptides significantly reduce blood pressure. However ACE inhibitory action in vivo, depends of the peptides degradation and intestinal absorption. During the digestion of whey proteins is released a mixture of different amino acid sequences, molecular weight, and different stability and permeability. The digestive conditions can alter the activity of the peptides, producing degraded sequences. Conversely, long-chain active peptides can present low tolerance to gastrointestinal digestion. Therefore, released peptides cannot be totally absorbed through the intestine to enter the blood circulation and exert systemic effects. In general whey peptides present low bioavailability and short in vivo half-lives (< 30 min).

Despite their susceptibility to metabolism, peptides are recognised as important therapeutic compounds and therefore several approaches have been designed to increase the oral delivery of peptides. This reason has promoted different research with the aim to obtain peptides with explicit properties by controlled digestion. Controlled reactions in the laboratory can increment the peptides activity, and produce whey peptides of short chains to obtain products with different properties of absorption and hydrophilicity. Today, there are an extensive research about the whey peptides to explore their potential in different formulations for improving health, using different alternatives of enhance (several publication are available in the literature).

As a concrete response to the observations from reviewer 1. Certainly the whey peptides are biomolecules with big molecular mass and high hydrophilicity; which result in a poor bioavailability, as it was already indicated. However, in recent times several peptides are studied to enhance their properties and oral route of delivery.

Thus, various approaches for achieving this purpose are now considered. The encapsulation is studied as an alternative to increment the bioavailability of peptides. The capsules are used as delivery systems, peptides protection against enzymatic degradation, to preserve the peptides activity, extend and control their release, and to enhance the characteristics of the peptides and thus, facilitate their uptake via cells.

According the objective of the research, the present manuscript describes the encapsulation of the antihypertensive peptides derived from the whey proteins hydrolysate and the analysis of the release of these peptides in vitro conditions. 

The effect of encapsulation on the preservation of the antihypertensive activity and controlled release of whey peptides were studied in simulated gastrointestinal digestion.

The results about these aspects are principally described in 3.3 section of this manuscript, and were also mentioned in the conclusions paragraph.    

Therefore, prepared capsules were not utilized for searching for new antihypertensive peptides.

Reviewer 2 Report

I confirmed the changes authors made. I suppose the manuscript has been improved enough.

Author Response

 There are no comments